# Spectral $k$-Support Norm Regularization

**Andrew M. McDonald, Massimiliano Pontil, Dimitris Stamos**
Department of Computer Science
University College London
{a.mcdonald,m.pontil,d.stamos}@cs.ucl.ac.uk

## Abstract

The $k$-support norm has successfully been applied to sparse vector prediction problems. We observe that it belongs to a wider class of norms, which we call the box-norms. Within this framework we derive an efficient algorithm to compute the proximity operator of the squared norm, improving upon the original method for the $k$-support norm. We extend the norms from the vector to the matrix setting and we introduce the spectral $k$-support norm. We study its properties and show that it is closely related to the multitask learning cluster norm. We apply the norms to real and synthetic matrix completion datasets. Our findings indicate that spectral $k$-support norm regularization gives state of the art performance, consistently improving over trace norm regularization and the matrix elastic net.

## 1   Introduction

In recent years there has been a great deal of interest in the problem of learning a low rank matrix from a set of linear measurements. A widely studied and successful instance of this problem arises in the context of matrix completion or collaborative filtering, in which we want to recover a low rank (or approximately low rank) matrix from a small sample of its entries, see e.g. [1, 2]. One prominent method to solve this problem is trace norm regularization: we look for a matrix which closely fits the observed entries and has a small trace norm (sum of singular values) [3, 4, 5]. Besides collaborative filtering, this problem has important applications ranging from multitask learning, to computer vision and natural language processing, to mention but a few.

In this paper, we propose new techniques to learn low rank matrices. These are inspired by the notion of the $k$-support norm [6], which was recently studied in the context of sparse vector prediction and shown to empirically outperform the Lasso [7] and Elastic Net [8] penalties. We note that this norm can naturally be extended to the matrix setting and its characteristic properties relating to the cardinality operator translate in a natural manner to matrices. Our approach is suggested by the observation that the $k$-support norm belongs to a broader class of norms, which makes it apparent that they can be extended to spectral matrix norms. Moreover, it provides a link between the spectral $k$-support norm and the *cluster norm*, a regularizer introduced in the context of multitask learning [9]. This result allows us to interpret the spectral $k$-support norm as a special case of the cluster norm and furthermore adds a new perspective of the cluster norm as a perturbation of the former.

The main contributions of this paper are threefold. First, we show that the $k$-support norm can be written as a parametrized infimum of quadratics, which we term the *box-norms*, and which are symmetric gauge functions. This allows us to extend the norms to orthogonally invariant matrix norms using a classical result by von Neumann [10]. Second, we show that the spectral box-norm is essentially equivalent to the cluster norm, which in turn can be interpreted as a perturbation of the spectral $k$-support norm, in the sense of the Moreau envelope [11]. Third, we use the infimum framework to compute the box-norm and the proximity operator of the squared norm in $\mathcal{O}(d \log d)$ time. Apart from improving on the $\mathcal{O}(d(k + \log d))$ algorithm in [6], this method allows one to use optimal first order optimization algorithms [12] with the cluster norm. Finally, we present numerical

experiments which indicate that the spectral $k$-support norm shows a significant improvement in performance over regularization with the trace norm and the matrix elastic net, on four popular matrix completion benchmarks.

The paper is organized as follows. In Section 2 we recall the $k$-support norm, and define the box-norm. In Section 3 we study their properties, we introduce the corresponding spectral norms, and we observe the connection to the cluster norm. In Section 4 we compute the norm and we derive a fast method to compute the proximity operator. Finally, in Section 5 we report on our numerical experiments. The supplementary material contains derivations of the results in the body of the paper.

## 2   Preliminaries

In this section, we recall the $k$-support norm and we introduce the box-norm and its dual. The $k$-support norm $\| \cdot \|_{(k)}$ was introduced in [6] as the norm whose unit ball is the convex hull of the set of vectors of cardinality at most $k$ and $\ell_2$-norm no greater than one. The authors show that the $k$-support norm can be written as the infimal convolution [11]

$$\|w\|_{(k)} = \inf \left\{ \sum_{g \in \mathcal{G}_k} \|v_g\|_2 : v_g \in \mathbb{R}^d, \; \text{supp}(v_g) \subseteq g, \; \sum_{g \in \mathcal{G}_k} v_g = w \right\}, \quad w \in \mathbb{R}^d, \qquad (1)$$

where $\mathcal{G}_k$ is the collection of all subsets of $\{1, \ldots, d\}$ containing at most $k$ elements, and for any $v \in \mathbb{R}^d$, the set $\text{supp}(v) = \{i : v_i \neq 0\}$ denotes the support of $v$. When used as a regularizer, the norm encourages vectors $w$ to be a sum of a limited number of vectors with small support. The $k$-support norm is a special case of the group lasso with overlap [13], where the cardinality of the support sets is at most $k$. Despite the complicated form of the primal norm, the dual norm has a simple formulation, namely the $\ell_2$-norm of the $k$ largest components of the vector

$$\|u\|_{*,(k)} = \sqrt{\sum_{i=1}^{k} (|u|_i^\downarrow)^2}, \quad u \in \mathbb{R}^d, \qquad (2)$$

where $|u|^\downarrow$ is the vector obtained from $u$ by reordering its components so that they are non-increasing in absolute value [6]. The $k$-support norm includes the $\ell_1$-norm and $\ell_2$-norm as special cases. This is clear from the dual norm since for $k = 1$ and $k = d$, it is equal to the $\ell_\infty$-norm and $\ell_2$-norm, respectively. We note that while definition (1) involves a combinatorial number of variables, [6] observed that the norm can be computed in $\mathcal{O}(d \log d)$.

We now define the box-norm, and in the following section we will show that the $k$-support norm is a special case of this family.

**Definition 2.1.** Let $0 \leq a \leq b$ and $c \in [ad, bd]$ and let $\Theta = \{\theta \in \mathbb{R}^d : a \leq \theta_i \leq b, \sum_{i=1}^d \theta_i \leq c\}$. The box-norm is defined as

$$\|w\|_\Theta = \sqrt{\inf_{\theta \in \Theta} \sum_{i=1}^{d} \frac{w_i^2}{\theta_i}}, \quad w \in \mathbb{R}^d. \qquad (3)$$

This formulation will be fundamental in deriving the proximity operator in Section 4.1. Note that we may assume without loss of generality that $b = 1$, as by rescaling we obtain an equivalent norm, however we do not explicitly fix $b$ in the sequel.

**Proposition 2.2.** *The norm* (3) *is well defined and the dual norm is* $\|u\|_{*,\Theta} = \sqrt{\sup_{\theta \in \Theta} \sum_{i=1}^{d} \theta_i u_i^2}$.

The result holds true in the more general case that $\Theta$ is a bounded convex subset of the strictly positive orthant (for related results see [14, 15, 16, 17, 18, 19] and references therein). In this paper we limit ourselves to the box constraints above. In particular we note that the constraints are invariant with respect to permutation of the components of $\Theta$, and as we shall see this property is key to extend the norm to matrices.

# 3 Properties of the Norms

In this section, we study the properties of the vector norms, and we extend the norms to the matrix setting. We begin by deriving the dual box-norm.

**Proposition 3.1.** *The dual box-norm is given by*

$$\|u\|_{*,\Theta} = \sqrt{a\|u\|_2^2 + (b-a)\|u\|_{*,(k)}^2 + (b-a)(\rho-k)(|u|_{k+1}^{\downarrow})^2}, \tag{4}$$

*where $\rho = \frac{c-da}{b-a}$ and $k$ is the largest integer not exceeding $\rho$.*

We see from (4) that the dual norm decomposes into two $\ell_2$-norms plus a residual term, which vanishes if $\rho = k$, and for the rest of this paper we assume this holds, which loses little generality.

Note that setting $a = 0, b = 1$, and $c = k \in \{1, \ldots, d\}$, the dual box-norm (4) is the $\ell_2$-norm of the largest $k$ components of $u$, and we recover the dual $k$-support norm in equation (2). It follows that the $k$-support norm is a box-norm with parameters $a = 0$, $b = 1$, $c = k$.

The following infimal convolution interpretation of the box-norm provides a link between the box-norm and the $k$-support norm, and illustrates the effect of the parameters.

**Proposition 3.2.** *If $0 < a \leq b$ and $c = (b-a)k + da$, for $k \in \{1, \ldots, d\}$, then*

$$\|w\|_{\Theta} = \inf \left\{ \sum_{g \in \mathcal{G}_k} \sqrt{\sum_{i \in g} \frac{v_{g,i}^2}{b} + \sum_{i \notin g} \frac{v_{g,i}^2}{a}} : v_g \in \mathbb{R}^d, \ \sum_{g \in \mathcal{G}_k} v_g = w \right\}. \tag{5}$$

Notice that if $b = 1$, then as $a$ tends to zero, we obtain the expression of the $k$-support norm (1), recovering in particular the support constraints. If $a$ is small and positive, the support constraints are not imposed, however effectively most of the weight for each $v_g$ tends to be concentrated on $\text{supp}(g)$. Hence, Proposition 3.2 suggests that the box-norm regularizer will encourage vectors $w$ whose dominant components are a subset of a union of a small number of groups $g \in \mathcal{G}_k$.

The previous results have characterized the $k$-support norm as a special case of the box-norm. Conversely, the box-norm can be seen as a perturbation of the $k$-support norm with a quadratic term.

**Proposition 3.3.** *Let $\|\cdot\|_{\Theta}$ be the box-norm on $\mathbb{R}^d$ with parameters $0 < a < b$ and $c = k(b-a)+da$, for $k \in \{1, \ldots, d\}$, then*

$$\|w\|_{\Theta}^2 = \min_{z \in \mathbb{R}^d} \left\{ \frac{1}{a}\|w - z\|_2^2 + \frac{1}{b-a}\|z\|_{(k)}^2 \right\}. \tag{6}$$

Consider the regularization problem $\min_{w \in \mathbb{R}^d} \|Xw - y\|_2^2 + \lambda\|w\|_{\Theta}^2$, with data $X$ and response $y$. Using Proposition 3.3 and setting $w = u + z$, we see that this problem is equivalent to

$$\min_{u,z \in \mathbb{R}^d} \left\{ \|X(u + z) - y\|_2^2 + \frac{\lambda}{a}\|u\|_2^2 + \frac{\lambda}{b-a}\|z\|_{(k)}^2 \right\}.$$

Furthermore, if $(\hat{u}, \hat{z})$ solves this problem then $\hat{w} = \hat{u} + \hat{z}$ solves problem (6). The solution $\hat{w}$ can therefore be interpreted as the superposition of a vector which has small $\ell_2$ norm, and a vector which has small $k$-support norm, with the parameter $a$ regulating these two components. Specifically, as $a$ tends to zero, in order to prevent the objective from blowing up, $\hat{u}$ must also tend to zero and we recover $k$-support norm regularization. Similarly, as $a$ tends to $b$, $\hat{z}$ vanishes and we have a simple ridge regression problem.

## 3.1 The Spectral $k$-Support Norm and the Spectral Box-Norm

We now turn our focus to the matrix norms. For this purpose, we recall that a norm $\|\cdot\|$ on $\mathbb{R}^{d \times m}$ is called orthogonally invariant if $\|W\| = \|UWV\|$, for any orthogonal matrices $U \in \mathbb{R}^{d \times d}$ and $V \in \mathbb{R}^{m \times m}$. A classical result by von Neumann [10] establishes that a norm is orthogonally invariant if and only if it is of the form $\|W\| = g(\sigma(W))$, where $\sigma(W)$ is the vector formed by the singular values of $W$ in nonincreasing order, and $g$ is a symmetric gauge function, that is a norm which is invariant under permutations and sign changes of the vector components.

**Lemma 3.4.** *If $\Theta$ is a convex bounded subset of the strictly positive orthant in $\mathbb{R}^d$ which is invariant under permutations, then $\|\cdot\|_\Theta$ is a symmetric gauge function.*

In particular, this readily applies to both the $k$-support norm and box-norm. We can therefore extend both norms to orthogonally invariant norms, which we term the spectral $k$-support norm and the spectral box-norm respectively, and which we write (with some abuse of notation) as $\|W\|_{(k)} = \|\sigma(W)\|_{(k)}$ and $\|W\|_\Theta = \|\sigma(W)\|_\Theta$. We note that since the $k$-support norm subsumes the $\ell_1$ and $\ell_2$-norms for $k = 1$ and $k = d$ respectively, the corresponding spectral $k$-support norms are equal to the trace and Frobenius norms respectively. We first characterize the unit ball of the spectral $k$-support norm.

**Proposition 3.5.** *The unit ball of the spectral $k$-support norm is the convex hull of the set of matrices of rank at most k and Frobenius norm no greater than one.*

Referring to the unit ball characterization of the $k$-support norm, we note that the restriction on the cardinality of the vectors whose convex hull defines the unit ball naturally extends to a restriction on the rank operator in the matrix setting. Furthermore, as noted in [6], regularization using the $k$-support norm encourages vectors to be sparse, but less so that the $\ell_1$-norm. In matrix problems, as the extreme points of the unit ball have rank $k$, Proposition 3.5 suggests that the spectral $k$-support norm for $k > 1$ should encourage matrices to have low rank, but less so than the trace norm.

## 3.2 Cluster Norm

We end this section by briefly discussing the cluster norm, which was introduced in [9] as a convex relaxation of a multitask clustering problem. The norm is defined, for every $W \in \mathbb{R}^{d \times m}$, as

$$\|W\|_{\text{cl}} = \sqrt{\inf_{S \in \mathcal{S}_m} \text{tr}(S^{-1} W^\top W)} \tag{7}$$

where $\mathcal{S}_m = \{S \in \mathbb{R}^{m \times m}, S \succeq 0 : aI \preceq S \preceq bI, \text{ tr} S = c\}$, and $0 < a \le b$. In [9] the authors state that the cluster norm of $W$ equals the box-norm of the vector formed by the singular values of $W$ where $c = (b-a)k+da$. Here we provide a proof of this result. Denote by $\lambda_i(\cdot)$ the eigenvalues of a matrix which we write in nonincreasing order $\lambda_1(\cdot) \ge \lambda_2(\cdot) \ge \cdots \ge \lambda_d(\cdot)$. Note that if $\theta_i$ are the eigenvalues of $S$ then $\theta_i = \lambda_{d-i+1}(S^{-1})$. We have that

$$\text{tr}(S^{-1} W^\top W) = \text{tr}(S^{-1} U \Sigma^2 U^\top) \ge \sum_{i=1}^m \lambda_{d-i+1}(S^{-1}) \lambda_i(W^\top W) = \sum_{i=1}^d \frac{\sigma_i^2(W)}{\theta_i}$$

where we have used the inequality [20, Sec. H.1.h] for $S^{-1}$, $W^\top W \succeq 0$. Since this inequality is attained whenever $S = U \text{Diag}(\theta) U$, where $U$ are the eigenvectors of $W^\top W$, we see that $\|W\|_{\text{cl}} = \|\sigma(W)\|_\Theta$, that is, the cluster norm coincides with the spectral box-norm. In particular, we see that the spectral $k$-support norm is a special case of the cluster norm, where we let $a$ tend to zero, $b = 1$ and $c = k$. Moreover, the methods to compute the norm and its proximity operator described in the following section can directly be applied to the cluster norm.

As in the case of the vector norm (Proposition 3.3), the spectral box-norm or cluster norm can be written as a perturbation of spectral $k$-support norm with a quadratic term.

**Proposition 3.6.** *Let $\|\cdot\|_\Theta$ be a matrix box-norm with parameters $a, b, c$ and let $k = \frac{c-da}{b-a}$. Then*

$$\|W\|_\Theta^2 = \min_Z \frac{1}{a} \|W - Z\|_F^2 + \frac{1}{b-a} \|Z\|_{(k)}^2.$$

In other words, this result shows that the cluster norm can be seen as the Moreau envelope [11] of a spectral $k$-support norm.

# 4 Computing the Norms and their Proximity Operator

In this section, we compute the norm and the proximity operator of the squared norm by explicitly solving the optimization problem in (3). We begin with the vector norm.

**Theorem 4.1.** *For every $w \in \mathbb{R}^d$ it holds that*

$$\|w\|_\Theta^2 = \frac{1}{b}\|w_Q\|_2^2 + \frac{1}{p}\|w_I\|_1^2 + \frac{1}{a}\|w_L\|_2^2, \tag{8}$$

*where $w_Q = (|w|_1^\downarrow, \ldots, |w|_q^\downarrow)$, $w_I = (|w|_{q+1}^\downarrow, \ldots, |w|_{d-\ell}^\downarrow)$, $w_L = (|w|_{d-\ell+1}^\downarrow, \ldots, |w|_d^\downarrow)$, and $q$ and $\ell$ are the unique integers in $\{0, \ldots, d\}$ that satisfy $q + \ell \leq d$,*

$$\frac{|w_q|}{b} \geq \frac{1}{p}\sum_{i=q+1}^{d-\ell}|w_i| > \frac{|w_{q+1}|}{b}, \quad \frac{|w_{d-\ell}|}{a} \geq \frac{1}{p}\sum_{i=q+1}^{d-\ell}|w_i| > \frac{|w_{d-\ell+1}|}{a}, \tag{9}$$

*$p = c - qb - \ell a$ and we have defined $|w_0| = \infty$ and $|w_{d+1}| = 0$.*

**Proof.** (Sketch) We need to solve the optimization problem

$$\inf_\theta \left\{ \sum_{i=1}^d \frac{w_i^2}{\theta_i} : a \leq \theta_i \leq b, \sum_{i=1}^d \theta_i \leq c \right\}. \tag{10}$$

We assume without loss of generality that the $w_i$ are ordered nonincreasing in absolute values, and it follows that at the optimum the $\theta_i$ are also ordered nonincreasing. We further assume that $w_i \neq 0$ for all $i$ and $c \leq db$, so the sum constraint will be tight at the optimum. The Lagrangian is given by

$$L(\theta, \alpha) = \sum_{i=1}^d \frac{w_i^2}{\theta_i} + \frac{1}{\alpha^2}\left(\sum_{i=1}^d \theta_i - c\right)$$

where $1/\alpha^2$ is a strictly positive multiplier to be chosen such that $S(\alpha) := \sum_{i=1}^d \theta_i(\alpha) = c$. We can then solve the original problem by minimizing the Lagrangian over the constraint $\theta \in [a, b]^d$. Due to the decoupling effect of the multiplier we can solve the simplified problem componentwise, obtaining the solution

$$\theta_i = \theta_i(\alpha) = \min(b, \max(a, \alpha|w_i|)) \tag{11}$$

where $S(\alpha) = c$. The minimizer has the form $\theta = (b, \ldots, b, \theta_{q+1}, \ldots, \theta_{d-\ell}, a, \ldots, a)$, where $q, \ell$ are determined by the value of $\alpha$. From $S(\alpha) = c$ we get $\alpha = p/(\sum_{i=q+1}^{d-\ell}|w_i|)$. The value of the norm in (8) follows by substituting $\theta$ into the objective. Finally, by construction we have $\theta_q \geq b > \theta_{q+1}$ and $\theta_{d-\ell} > a \geq \theta_{d-\ell+1}$, which give rise to the conditions in (9). ■

Theorem 4.1 suggests two methods for computing the box-norm. First we find $\alpha$ such that $S(\alpha) = c$; this value uniquely determines $\theta$ in (11), and the norm follows by substitution into (10). Alternatively, we identify $q$ and $\ell$ that jointly satisfy (9) and we compute the norm using (8). Taking advantage of the structure of $\theta$ in the former method leads to a computation time that is $\mathcal{O}(d \log d)$.

**Theorem 4.2.** *The computation of the box-norm can be completed in $\mathcal{O}(d \log d)$ time.*

The $k$-support norm is a special case of the box-norm, and as a direct corollary of Theorem 4.1 and Theorem 4.2, we recover [6, Proposition 2.1].

## 4.1 Proximity Operator

Proximal gradient methods can be used to solve optimization problems of the form $\min_w f(w) + \lambda g(w)$, where $f$ is a convex loss function with Lipschitz continuous gradient, $\lambda > 0$ is a regularization parameter, and $g$ is a convex function for which the proximity operator can be computed efficiently, see [12, 21, 22] and references therein. The proximity operator of $g$ with parameter $\rho > 0$ is defined as

$$\mathrm{prox}_{\rho g}(w) = \mathrm{argmin}\left\{\frac{1}{2}\|x - w\|^2 + \rho g(x) : x \in \mathbb{R}^d\right\}.$$

We now use the infimum formulation of the box-norm to derive the proximity operator of the squared norm.

---

**Algorithm 1** Computation of $x = \text{prox}_{\frac{\lambda}{2}\|\cdot\|_{\ominus}^2}(w)$.

---

**Require:** parameters $a, b, c, \lambda$.

   **1.** Sort points $\{\alpha^i\}_{i=1}^{2d} = \left\{\frac{a+\lambda}{|w_j|}, \frac{b+\lambda}{|w_j|}\right\}_{j=1}^{d}$ such that $\alpha^i \leq \alpha^{i+1}$;
   **2.** Identify points $\alpha^i$ and $\alpha^{i+1}$ such that $S(\alpha^i) \leq c$ and $S(\alpha^{i+1}) \geq c$ by binary search;
   **3.** Find $\alpha^*$ between $\alpha^i$ and $\alpha^{i+1}$ such that $S(\alpha^*) = c$ by linear interpolation;
   **4.** Compute $\theta_i(\alpha^*)$ for $i = 1, \ldots, d$;
   **5.** Return $x_i = \frac{\theta_i w_i}{\theta_i + \lambda}$ for $i = 1, \ldots, d$.

---

**Theorem 4.3.** *The proximity operator of the square of the box-norm at point $w \in \mathbb{R}^d$ with parameter $\frac{\lambda}{2}$ is given by $\text{prox}_{\frac{\lambda}{2}\|\cdot\|_{\ominus}^2}(w) = (\frac{\theta_1 w_1}{\theta_1 + \lambda}, \ldots, \frac{\theta_d w_d}{\theta_d + \lambda})$, where*

$$\theta_i = \theta_i(\alpha) = \min(b, \max(a, \alpha|w_i| - \lambda)) \tag{12}$$

*and $\alpha$ is chosen such that $S(\alpha) := \sum_{i=1}^{d} \theta_i(\alpha) = c$. Furthermore, the computation of the proximity operator can be completed in $\mathcal{O}(d \log d)$ time.*

The proof follows a similar reasoning to the proof of Theorem 4.1. Algorithm 1 illustrates the computation of the proximity operator for the squared box-norm in $\mathcal{O}(d \log d)$ time. This includes the $k$-support as a special case, where we let $a$ tend to zero, and set $b = 1$ and $c = k$, which improves upon the complexity of the $\mathcal{O}(d(k + \log d))$ computation provided in [6], and we illustrate the improvement empirically in Table 1.

### 4.2 Proximity Operator for Orthogonally Invariant Norms

The computational considerations outlined above can be naturally extended to the matrix setting by using von Neumann's trace inequality (see, e.g. [23]). Here we comment on the computation of the proximity operator, which is important for our numerical experiments in the following section. The proximity operator of an orthogonally invariant norm $\|\cdot\| = g(\sigma(\cdot))$ is given by

$$\text{prox}_{\|\cdot\|}(W) = U\text{diag}(\text{prox}_g(\sigma(W)))V^\top, \quad W \in \mathbb{R}^{m \times d},$$

where $U$ and $V$ are the matrices formed by the left and right singular vectors of $W$ (see e.g. [24, Prop 3.1]). Using this result we can employ proximal gradient methods to solve matrix regularization problems using the squared spectral $k$-support norm and spectral box-norm.

## 5 Numerical Experiments

In this section, we report on the statistical performance of the spectral regularizers in matrix completion experiments. We also offer an interpretation of the role of the parameters in the box-norm and we empirically verify the improved performance of the proximity operator computation (see Table 1). We compare the trace norm (*tr*) [25], matrix elastic net (*en*) [26], spectral $k$-support (*ks*) and the spectral box-norm (*box*). The Frobenius norm, which is equal to the spectral $k$-support norm for $k = d$, performed considerably worse than the trace norm and we omit the results here. We report test error and standard deviation, matrix rank ($r$) and optimal parameter values for $k$ and $a$, which were determined by validation, as were the regularization parameters. When comparing performance, we used a *t-test* to determine statistical significance at a level of $p < 0.001$. For the optimization we used an accelerated proximal gradient method (FISTA), see e.g. [12, 21, 22], with the percentage change in objective as convergence criterion, with a tolerance of $10^{-5}$ for the simulated datasets and $10^{-3}$ for the real datasets. As is typical with spectral regularizers we found that the spectrum of the learned matrix exhibited a rapid decay to zero. In order to explicitly impose a low rank on the solution we included a final step where we hard-threshold the singular values of the final matrix below a level determined by validation. We report on both sets of results below.

### 5.1 Simulated Data

**Matrix Completion.** We applied the norms to matrix completion on noisy observations of low rank matrices. Each $m \times m$ matrix was generated as $W = AB^\top + E$, where $A, B \in \mathbb{R}^{m \times r}$, $r \ll m$, and

Table 1: Comparison of proximity operator algorithms for the $k$-support norm (time in s), $k = 0.05d$. Algorithm 1 is the method in [6], Algorithm 2 is our method.

| $d$ | 1,000 | 2,000 | 4,000 | 8,000 | 16,000 | 32,000 |
|---|---|---|---|---|---|---|
| Alg. 1 | 0.0443 | 0.1567 | 0.5907 | 2.3065 | 9.0080 | 35.6199 |
| Alg. 2 | 0.0011 | 0.0016 | 0.0026 | 0.0046 | 0.0101 | 0.0181 |

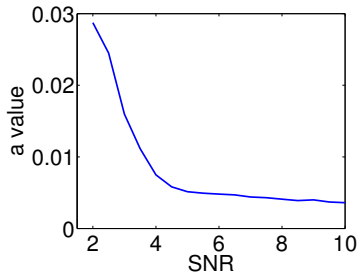

Figure 1: Impact of signal to noise on $a$.

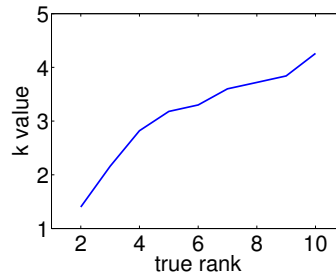

Figure 2: Impact of matrix rank on $k$.

the entries of $A$, $B$ and $E$ are i.i.d. standard Gaussian. We set $m = 100$, $r \in \{5, 10\}$ and we sampled uniformly a percentage $\rho \in \{10\%, 20\%, 30\%\}$ of the entries for training, and used a fixed 10% for validation. The error was measured as $\|\text{true} - \text{predicted}\|^2 / \|\text{true}\|^2$ [5] and averaged over 100 trials. The results are summarized in Table 2. In the thresholding case, all methods recovered the rank of the true noiseless matrix. The spectral box-norm generated the lowest test errors in all regimes, with the spectral $k$-support a close second, in particular in the thresholding case. This suggests that the non zero parameter $a$ in the spectral box-norm counteracted the noise to some extent.

**Role of Parameters.** In the same setting we investigated the role of the parameters in the box-norm. As previously discussed, parameter $b$ can be set to 1 without loss of generality. Figure 1 shows the optimal value of $a$ chosen by validation for varying signal to noise ratios (SNR), keeping $k$ fixed. We see that for greater noise levels (smaller SNR), the optimal value for $a$ increases. While for $a > 0$, the recovered solutions are not sparse, as we show below this can still lead to improved performance in experiments, in particular in the presence of noise. Figure 2 shows the optimal value of $k$ chosen by validation for matrices with increasing rank, keeping $a$ fixed. We notice that as the rank of the matrix increases, the optimal $k$ value increases, which is expected since it is an upper bound on the sum of the singular values.

Table 2: Matrix completion on simulated data sets, without (left) and with (right) thresholding.

| dataset | norm | test error | $r$ | $k$ | $a$ | dataset | norm | test error | $r$ | $k$ | $a$ |
|---|---|---|---|---|---|---|---|---|---|---|---|
| rank 5 | tr | 0.8184 (0.03) | 20 | - | - | rank 5 | tr | 0.7799 (0.04) | 5 | - | - |
| $\rho=10\%$ | en | 0.8164 (0.03) | 20 | - | - | $\rho=10\%$ | en | 0.7794 (0.04) | 5 | - | - |
| | ks | 0.8036 (0.03) | 16 | 3.6 | - | | ks | 0.7728 (0.04) | 5 | 4.23 | - |
| | box | 0.7805 (0.03) | 87 | 2.9 | 1.7e-2 | | box | 0.7649 (0.04) | 5 | 3.63 | 8.1e-3 |
| rank 5 | tr | 0.4085 (0.03) | 23 | - | - | rank 5 | tr | 0.3449 (0.02) | 5 | - | - |
| $\rho=20\%$ | en | 0.4081 (0.03) | 23 | - | - | $\rho=20\%$ | en | 0.3445 (0.02) | 5 | - | - |
| | ks | 0.4031 (0.03) | 21 | 3.1 | - | | ks | 0.3381 (0.02) | 5 | 2.97 | - |
| | box | 0.3898 (0.03) | 100 | 1.3 | 9e-3 | | box | 0.3380 (0.02) | 5 | 3.28 | 1.9e-3 |
| rank 10 | tr | 0.6356 (0.03) | 27 | - | - | rank 10 | tr | 0.6084 (0.03) | 10 | - | - |
| $\rho=20\%$ | en | 0.6359 (0.03) | 27 | - | - | $\rho=20\%$ | en | 0.6074 (0.03) | 10 | - | - |
| | ks | 0.6284 (0.03) | 24 | 4.4 | - | | ks | 0.6000 (0.03) | 10 | 5.02 | - |
| | box | 0.6243 (0.03) | 89 | 1.8 | 9e-3 | | box | 0.6000 (0.03) | 10 | 5.22 | 1.9e-3 |
| rank 10 | tr | 0.3642 (0.02) | 36 | - | - | rank 10 | tr | 0.3086 (0.02) | 10 | - | - |
| $\rho=30\%$ | en | 0.3638 (0.002) | 36 | - | - | $\rho=30\%$ | en | 0.3082 (0.02) | 10 | - | - |
| | ks | 0.3579 (0.02) | 33 | 5.0 | - | | ks | 0.3025 (0.02) | 10 | 5.13 | - |
| | box | 0.3486 (0.02) | 100 | 2.5 | 9e-3 | | box | 0.3025 (0.02) | 10 | 5.16 | 3e-4 |

Table 3: Matrix completion on real data sets, without (left) and with (right) thresholding.

| dataset | norm | test error | $r$ | $k$ | $a$ | dataset | norm | test error | $r$ | $k$ | $a$ |
|---|---|---|---|---|---|---|---|---|---|---|---|
| MovieLens | tr | 0.2034 | 87 | - | - | MovieLens | tr | 0.2017 | 13 | - | - |
| 100k | en | 0.2034 | 87 | - | - | 100k | en | 0.2017 | 13 | - | - |
| $\rho = 50\%$ | ks | 0.2031 | 102 | 1.00 | - | $\rho = 50\%$ | ks | 0.1990 | 9 | 1.87 | - |
| | box | 0.2035 | 943 | 1.00 | 1e-5 | | box | 0.1989 | 10 | 2.00 | 1e-5 |
| MovieLens | tr | 0.1821 | 325 | - | - | MovieLens | tr | 0.1790 | 17 | - | - |
| 1M | en | 0.1821 | 319 | - | - | 1M | en | 0.1789 | 17 | - | - |
| $\rho = 50\%$ | ks | 0.1820 | 317 | 1.00 | - | $\rho = 50\%$ | ks | 0.1782 | 17 | 1.80 | - |
| | box | 0.1817 | 3576 | 1.09 | 3e-5 | | box | 0.1777 | 19 | 2.00 | 1e-6 |
| Jester 1 | tr | 0.1787 | 98 | - | - | Jester 1 | tr | 0.1752 | 11 | - | - |
| 20 per line | en | 0.1787 | 98 | - | - | 20 per line | en | 0.1752 | 11 | - | - |
| | ks | 0.1764 | 84 | 5.00 | - | | ks | 0.1739 | 11 | 6.38 | - |
| | box | 0.1766 | 100 | 4.00 | 1e-6 | | box | 0.1726 | 11 | 6.40 | 2e-5 |
| Jester 3 | tr | 0.1988 | 49 | - | - | Jester 3 | tr | 0.1959 | 3 | - | - |
| 8 per line | en | 0.1988 | 49 | - | - | 8 per line | en | 0.1959 | 3 | - | - |
| | ks | 0.1970 | 46 | 3.70 | - | | ks | 0.1942 | 3 | 2.13 | - |
| | box | 0.1973 | 100 | 5.91 | 1e-3 | | box | 0.1940 | 3 | 4.00 | 8e-4 |

## 5.2 Real Data

**Matrix Completion (MovieLens and Jester).** In this section we report on matrix completion on real data sets. We observe a percentage of the (user, rating) entries of a matrix and the task is to predict the unobserved ratings, with the assumption that the true matrix has low rank. The datasets we considered were MovieLens 100k and MovieLens 1M (*http://grouplens.org/datasets/movielens/*), which consist of user ratings of movies, and Jester 1 and Jester 3 (*http://goldberg.berkeley.edu/jester-data/*), which consist of users and ratings of jokes (Jester 2 showed essentially identical performance to Jester 1). Following [4], for MovieLens we uniformly sampled $\rho = 50\%$ of the available entries for each user for training, and for Jester 1 and Jester 3 we sampled 20, respectively 8, ratings per user, and we used 10% for validation. The error was measured as normalized mean absolute error, $\frac{\|\text{true} - \text{predicted}\|^2}{\#\text{observations}/(r_{\max} - r_{\min})}$, where $r_{\min}$ and $r_{\max}$ are lower and upper bounds for the ratings [4]. The results are outlined in Table 3. In the thresholding case, the spectral box and $k$-support norms had the best performance. In the absence of thresholding, the spectral $k$-support showed slightly better performance. Comparing to the synthetic data sets, this suggests that in the absence of noise the parameter $a$ did not provide any benefit. We note that in the absence of thresholding our results for the trace norm on MovieLens 100k agreed with those in [3].

## 6 Conclusion

We showed that the $k$-support norm belongs to the family of box-norms and noted that these can be naturally extended from the vector to the matrix setting. We also provided a connection between the $k$-support norm and the cluster norm, which essentially coincides with the spectral box-norm. We further observed that the cluster norm is a perturbation of the spectral $k$-support norm, and we were able to compute the norm and its proximity operator. Our experiments indicate that the spectral box-norm and $k$-support norm consistently outperform the trace norm and the matrix elastic net on various matrix completion problems. With a single parameter to validate, compared to two for the spectral box-norm, our results suggest that the spectral $k$-support norm is a powerful alternative to the trace norm and the elastic net, which has the same number of parameters. In future work, we would like to study the application of the norms to clustering problems in multitask learning [9], in particular the impact of centering. It would also be valuable to derive statistical inequalities and Rademacher complexities for these norms.

## Acknowledgements

We would like to thank Andreas Maurer, Charles Micchelli and especially Andreas Argyriou for useful discussions. Part of this work was supported by EPSRC Grant EP/H027203/1.

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
