[Supplementary Material]

## Supplementary Material

In this appendix, we collect some auxiliary results and we provide proofs of the results stated in the main body of the paper.

## A    Auxiliary Results

Recall that a subset $A$ of a real vector space $X$ is called *balanced* if $\alpha A \subset A$ whenever $|\alpha| \leq 1$. Furthermore, $A$ is called *absorbing* if for any $x \in X$, $x \in \lambda A$ for some $\lambda(x) > 0$. For a proof of the following lemma see e.g. [27, §1.35].

**Lemma A.1.** *Let $X$ be a real vector space and let $A \subset X$ be a convex, balanced, and absorbing set. The Minkowski functional $\mu_A$ of $A$, given, for every $x \in X$, by the formula*

$$\mu_A(x) = \inf\{\lambda > 0 : x \in \lambda A\}$$

*defines a seminorm on $X$. In addition, if $\mu_A(x) > 0$ for every $x \neq 0$, then $\mu_A$ defines a norm on $X$.*

The next result is due to von Neumann [10], see also [23].

**Theorem A.2** (Von Neumann's trace inequality)**.** *For any $d \times m$ matrices $X$ and $Y$,*

$$\mathrm{tr}(XY^\top) \leq \langle \sigma(X), \sigma(Y) \rangle.$$

*Equality holds if and only if $X$ and $Y$ admit a simultaneous singular value decomposition, that is*

$$X = U\mathrm{diag}(\sigma(X))V^\top, \qquad Y = U\mathrm{diag}(\sigma(Y))V^\top,$$

*where $U \in \mathbb{R}^{d \times d}$ and $V \in \mathbb{R}^{m \times m}$ are orthogonal matrices.*

The following result, which is presented in [6, Section 2] is key for the proof of Theorem 3.5.

**Proposition A.3.** *The unit ball of the vector $k$-support norm is equal to the convex hull of the set $\{w \in \mathbb{R}^d : \mathrm{card}(w) \leq k, \|w\|_2 \leq 1\}$.*

Theorems 4.1 and 4.3 make use of the following result, which follows from [17], Theorem 3.1.

**Lemma A.4.** *Let $w \in \mathbb{R}, \beta > 0$, and define $g(\theta) = \frac{w^2}{\theta} + \beta^2\theta(\theta > 0)$. For $0 < a \leq b$, the unique solution to the problem $\min\{g(\theta) : a \leq \theta \leq b\}$ is given by*

$$\theta = \begin{cases} a, & \text{if } \frac{|w|}{\beta} < a, \\ \frac{|w|}{\beta}, & \text{if } a \leq \frac{|w|}{\beta} \leq b, \\ b, & \text{if } \frac{|w|}{\beta} > b. \end{cases}$$

**Proof.** For fixed $w$, the objective function is strictly convex on $\mathbb{R}^d_{++}$ and has a unique minimum on $(0, \infty)$ (see Figure 1.b in [17] for a one-dimensional illustration). The derivative of the objective function is zero for $\theta = \theta^* := |w|/\beta$, strictly positive below $\theta^*$ and strictly increasing above $\theta^*$. Considering these three cases we recover the expression in statement of the lemma. ∎

## B    Proofs

**Proof of Proposition 2.2.** Consider the expression for the dual norm. The function $\|\cdot\|_\Theta$ is a norm since it is a supremum of norms. Recall that the Fenchel conjugate $h^*$ of a function $h : \mathbb{R}^d \to \mathbb{R}$ is defined for every $u \in \mathbb{R}^d$ as $h^*(u) = \sup\{\langle u, w\rangle - h(w) : w \in \mathbb{R}^d\}$. It is a standard result from convex analysis that for any norm $\|\cdot\|$, the Fenchel conjugate of the function $h := \frac{1}{2}\|\cdot\|^2$ satisfies

$h^* = \frac{1}{2} \| \cdot \|_*^2$, where $\| \cdot \|_*$ is the corresponding dual norm (see, e.g. [23]). By the same result, for any norm the biconjugate is equal to the norm, that is $(\| \cdot \|^*)^* = \| \cdot \|$. Applying this to the dual norm we have, for every $w \in \mathbb{R}^d$,

$$h(w) = \sup_{u \in \mathbb{R}^d} \{\langle w, u \rangle - h^*(u)\} = \sup_{u \in \mathbb{R}^d} \inf_{\theta \in \Theta} \left\{ \sum_{i=1}^d \left( w_i u_i - \frac{1}{2} \theta_i u_i^2 \right) \right\}.$$

This is a minimax problem in the sense of von Neumann [28], and we can exchange the order of the inf and the sup, and solve the latter (which is in fact a maximum) componentwise. The gradient with respect to $u_i$ is zero for $u_i = \frac{w_i}{\theta_i}$, and substituting this into the objective we get

$$h(w) = \frac{1}{2} \inf_{\theta \in \Theta} \sum_{i=1}^d \frac{w_i^2}{\theta_i}.$$

It follows that the infimum expression in (3) defines a norm, and the two norms are duals of each other as required. ∎

**Proof of Proposition 3.1.** We make the change of variable $\phi_i = \frac{\theta_i - a}{b - a}$ and observe that the constraints on $\theta$ induce the constraint set $\{\phi \in (0, 1]^d, \ \sum_{i=1}^d \phi_i \leq \rho\}$, where $\rho = \frac{c - da}{b - a}$. Furthermore

$$\sum_{i=1}^d \theta_i u_i^2 = a \|u\|_2^2 + (b - a) \sum_{i=1}^d \phi_i u_i^2.$$

The result then follows by taking the supremum over $\phi$. ∎

**Proof of Proposition 3.2.** Equation 5 defines a norm and we will show that its norm coincides with the dual of the $\Theta$-norm given by equation (4). To simplify the exposition we define the norm

$$\|v\|_g^2 = \sum_{i \in g} \frac{v_i^2}{b} + \sum_{i \notin g} \frac{v_i^2}{a}, \quad v \in \mathbb{R}^d,$$

whose corresponding dual norm is

$$\|u\|_{*,g}^2 = b \sum_{i \in g} u_i^2 + a \sum_{i \notin g} u_i^2, \quad u \in \mathbb{R}^d.$$

Furthermore for every $u \in \mathbb{R}^d$ and $g \subseteq \{1, \ldots, d\}$, we define the vectors $u_{|g} = (u_i I_{\{i \in g\}})_{i=1}^d$ and $u_{|g^c} = (u_i I_{\{i \notin g\}})_{i=1}^d$.

We have, for every $u \in \mathbb{R}^d$, $u \neq 0$, that

$$\sup_{w \in \mathbb{R}^d} \frac{\langle w, u \rangle}{\|w\|} = \sup_{\{v_g\}} \frac{\sum_{g \in \mathcal{G}_k} \langle v_g, u \rangle}{\sum_{g \in \mathcal{G}_k} \|v_g\|_g}$$

$$\leq \sup_{\{v_g\}} \frac{\sum_{g \in \mathcal{G}_k} \|v_g\|_g \|u\|_{*,g}}{\sum_{g \in \mathcal{G}_k} \|v_g\|_g}$$

$$\leq \max_{g \in \mathcal{G}_k} \|u\|_{*,g}, \tag{13}$$

where we have used Cauchy-Schwarz and Hölder inequalities. We can make the first inequality tight by setting $v_g = \lambda_g (b u_{|g} + a u_{|g^c})$ and the second inequality tight by requiring $\lambda_g = 0$ whenever $g \notin \operatorname{argmax}_{g' \in \mathcal{G}_k} \|u\|_{*,g'}$, see e.g. [29, Sects. 5.4.14. and 5.4.15]. Note that the right hand side in (13) is maximized when $g = \{i_1, \ldots, i_k\}$ such that $|u_{i_1}| \geq \cdots |u_{i_k}|$ and the expression coincides with (4) for $\rho = k$. ∎

**Proof of Proposition 3.3.** Consider the definition of the norm $\|w\|_\Theta$ in (3). We make the change of variables $\phi_i = \frac{\theta_i - a}{b-a}$, and write

$$\|w\|_\Theta^2 = \min_{\theta \in \Theta} \sum_{i=1}^d \frac{w_i^2}{\theta_i} = \frac{\gamma}{a} \min_{\phi \in \Phi} \sum_{i=1}^d \frac{w_i^2}{\phi_i + \gamma}, \tag{14}$$

where we have defined $\gamma = \frac{a}{b-a}$ and $\Phi = \{\phi \in (0,1]^d : \sum_{i=1}^d \phi_i \le k\}$. We observe that

$$\min_{z \in \mathbb{R}^d} \left\{ \|w - z\|_2^2 + \gamma \|z\|_\Phi^2 \right\} = \min_{z \in \mathbb{R}^d} \min_{\phi \in \Phi} \left\{ \sum_{i=1}^d (w_i - z_i)^2 + \gamma \frac{z_i^2}{\phi_i} \right\} = \gamma \min_{\phi \in \Phi} \sum_{i=1}^d \frac{w_i^2}{\phi_i + \gamma}, \tag{15}$$

where we have interchanged the order of the minimization problems and solved for $z_i$ component-wise. The result follows by combining equations (14) and (15). ∎

**Proof of Lemma 3.4.** Let $g(w) = \|w\|_\Theta$. We need to show that $g$ is a norm which is invariant under permutations and sign changes. By Proposition 2.2, $g$ is a norm, so it remains to show that $g(w_1, ..., w_d) = g(w_{\pi(1)}, \ldots, w_{\pi(d)})$ for every permutation $\pi$, and $g(Jw) = g(w)$ for every diagonal matrix $J$ with entries $\pm 1$. The latter property is immediate. The former property follows since the set $\Theta$-norm is permutation invariant. ∎

**Proof of Proposition 3.5.** For any $W \in \mathbb{R}^{d \times m}$, define the following sets

$$T_k = \{W \in \mathbb{R}^{d \times m} : \mathrm{rank}(W) \le k, \|W\|_F \le 1\}, \quad A_k = \mathrm{co}(T_k),$$

and consider the following functional

$$\lambda(W) = \inf\{\lambda > 0 : W \in \lambda A_k\}, \quad W \in \mathbb{R}^{d \times m}. \tag{16}$$

By Lemma A.1, $\lambda$ defines a norm on $\mathbb{R}^{d \times m}$ with unit ball equal to $A_k$. Since the constraints in $T_k$ involve spectral functions, the sets $T_k$ and $A_k$ are invariant to left and right multiplication by orthogonal matrices. It follows that $\lambda$ is a spectral function, that is $\lambda(W)$ is defined in terms of the singular values of $W$, and by von Neumann's Theorem [10] the norm it defines is orthogonally invariant and we have

$$\lambda(W) = \inf\{\lambda > 0 : W \in \lambda A_k\}$$

$$= \inf\{\lambda > 0 : \sigma(W) \in \lambda C_k\}$$

$$= \|\sigma(W)\|_{(k)},$$

where we have defined the set $C_k = \mathrm{co}\{w \in \mathbb{R}^d : \|w\|_2 \le 1, \; \mathrm{card}(w) \le k\}$ and we have used the fact that the unit ball of the $k$-support norm is the convex hull of $C_k$ [6, Section 2] in the penultimate step. It follows that the norm defined by (16) is the spectral $k$-support norm. ∎

**Proof of Proposition 3.6.** By von Neumann's trace inequality (Theorem A.2) we have

$$\frac{1}{a}\|W - Z\|_F^2 + \frac{1}{b-a}\|Z\|_{(k)}^2 = \frac{1}{a}\left(\|W\|_F^2 + \|Z\|_F^2 - 2\langle W, Z \rangle\right) + \frac{1}{b-a}\|Z\|_{(k)}^2$$

$$\ge \frac{1}{a}\left(\|\sigma(W)\|_2^2 + \|\sigma(Z)\|_2^2 - 2\langle \sigma(W), \sigma(Z) \rangle\right) + \frac{1}{b-a}\|\sigma(Z)\|_{(k)}^2$$

$$= \frac{1}{a}\|\sigma(W) - \sigma(Z)\|_2^2 + \frac{1}{b-a}\|\sigma(Z)\|_{(k)}^2.$$

Furthermore the inequality is tight if $W$ and $Z$ have the same ordered set of singular vectors. Hence

$$\min_{Z \in \mathbb{R}^{d \times m}} \left\{ \frac{1}{a} \|W - Z\|_F^2 + \frac{1}{b-a} \|Z\|_{(k)}^2 \right\} = \min_{z \in \mathbb{R}^d} \left\{ \frac{1}{a} \|\sigma(W) - z\|_2^2 + \frac{1}{b-a} \|z\|_{(k)}^2 \right\} = \|\sigma(W)\|_{(k)}^2,$$

where the last equality follows by Proposition 3.3 ∎

**Proof of Theorem 4.1.** We solve the constrained optimization problem

$$\inf \left\{ \sum_{i=1}^{d} \frac{w_i^2}{\theta_i} : a \le \theta_i \le b, \sum_{i=1}^{d} \theta_i \le c \right\}. \tag{17}$$

To simplify notation we assume without loss of generality that $w_i$ are positive and ordered nonincreasing, and note that the optimal $\theta_i$ are ordered nonincreasing. To see this, let $\theta^* = \operatorname{argmin}_{\theta \in \Theta} \sum_{i=1}^{d} \frac{w_i^2}{\theta_i}$. Now suppose that $\theta_i^* < \theta_j^*$ for some $i < j$ and define $\hat{\theta}$ to be identical to $\theta^*$, except with the $i$ and $j$ elements exchanged. The difference in objective values is

$$\sum_{i=1}^{d} \frac{w_i^2}{\hat{\theta}_i} - \sum_{i=1}^{d} \frac{w_i^2}{\theta_i^*} = (w_i^2 - w_j^2) \left( \frac{1}{\theta_j^*} - \frac{1}{\theta_i^*} \right),$$

which is negative so $\theta^*$ cannot be a minimizer.

We further assume without loss of generality that $w_i \ne 0$ for all $i$, and $c \le db$ (see Remark B.1 below). The objective is continuous and we take the infimum over a closed bounded set, so a solution exists, the solution is a minimum, and it is unique by strict convexity. Furthermore, since $c \le db$, the sum constraint will be tight at the optimum.

Consider the Lagrangian function

$$L(\theta, \alpha) = \sum_{i=1}^{d} \frac{w_i^2}{\theta_i} + \frac{1}{\alpha^2} \left( \sum_{i=1}^{d} \theta_i - c \right), \tag{18}$$

where $1/\alpha^2$ is a strictly positive multiplier, and $\alpha$ is to be chosen to make the sum constraint tight, call this value $\alpha^*$. Let $\theta^*$ be the minimizer of $L(\theta, \alpha^*)$ over $\theta$ subject to $a \le \theta_i \le b$.

We claim that $\theta^*$ solves equation (17). Indeed, for any $\theta \in [a,b]^d$, $L(\theta^*, \alpha^*) \le L(\theta, \alpha^*)$, which implies that

$$\sum_{i=1}^{d} \frac{w_i^2}{\theta_i^*} \le \sum_{i=1}^{d} \frac{w_i^2}{\theta_i} + \frac{1}{(\alpha^*)^2} \left( \sum_{i=1}^{d} \theta_i - c \right).$$

If in addition we impose the constraint $\sum_{i=1}^{d} \theta_i \le c$, the second term on the right hand side is at most zero, so we have for all such $\theta$

$$\sum_{i=1}^{d} \frac{w_i^2}{\theta_i^*} \le \sum_{i=1}^{d} \frac{w_i^2}{\theta_i},$$

whence it follows that $\theta^*$ is the minimizer of (17).

We can therefore solve the original problem by minimizing the Lagrangian (18) over the box constraint. Due to the coupling effect of the multiplier, the problem is separable, and we can solve the simplified problem componentwise using Lemma A.4. It follows that

$$\theta_i = \begin{cases} a, & \text{if} \quad \alpha < \frac{a}{|w_i|}, \\ \alpha |w_i|, & \text{if} \quad \frac{a}{|w_i|} \le \alpha \le \frac{b}{|w_i|}, \\ b, & \text{if} \quad \alpha > \frac{b}{|w_i|}, \end{cases}$$

where $\alpha > 0$ is such that $\sum_{i=1}^{d} \theta_i(\alpha) = c$. Note also that in the main body of the paper we use the equivalent compact notation $\theta_i = \theta_i(\alpha) = \min(b, \max(a, \alpha|w_i|))$.

The minimizer then has the form

$$\theta = (\underbrace{b, \ldots, b}_{q}, \theta_{q+1}, \ldots, \theta_{d-\ell}, \underbrace{a, \ldots, a}_{\ell}),$$

where $q, \ell \in \{0, \ldots, d\}$ are determined by the value of $\alpha$ which satisfies

$$S(\alpha) = \sum_{i=1}^{d} \theta_i(\alpha) = qb + \sum_{i=q+1}^{d-\ell} \alpha|w_i| + \ell a = c,$$

i.e. $\alpha = p/\left(\sum_{i=q+1}^{d-\ell} |w_i|\right)$, where $p = c - qb - \ell a$.

The value of the norm follows by substituting $\theta$ into the objective and we get

$$\|w\|_{\Theta}^2 = \sum_{i=1}^{q} \frac{|w_i|^2}{b} + \frac{1}{p}\left(\sum_{i=q+1}^{d-\ell} |w_i|\right)^2 + \sum_{i=d-\ell+1}^{d} \frac{|w_i|^2}{a}$$

$$= \frac{1}{b}\|w_Q\|_2^2 + \frac{1}{p}\|w_I\|_1^2 + \frac{1}{a}\|w_L\|_2^2,$$

as required. We can further characterize $q$ and $\ell$ by considering the form of $\theta_i$. By construction we have $\theta_q \geq b > \theta_{q+1}$ and $\theta_{d-\ell} > a \geq \theta_{d-\ell+1}$, or equivalently

$$\frac{|w_q|}{b} \geq \frac{1}{p}\sum_{i=q+1}^{d-\ell} |w_i| > \frac{|w_{q+1}|}{b}, \text{ and}$$

$$\frac{|w_{d-\ell}|}{a} \geq \frac{1}{p}\sum_{i=q+1}^{d-\ell} |w_i| > \frac{|w_{d-\ell+1}|}{a},$$

and we are done. ∎

**Remark B.1.** The case where some $w_i$ are zero follows from the case that we have considered in the theorem. If $w_i = 0$ for $n < i \leq d$, then clearly we must have $\theta_i = a$ for all such $i$. We then consider the $n$-dimensional problem of finding $(\theta_1, \ldots, \theta_n)$ that minimizes $\sum_{i=1}^{n} \frac{w_i^2}{\theta_i}$, subject to $a \leq \theta_i \leq b$, and $\sum_{i=1}^{n} \theta_i \leq c'$, where $c' = c - (d-n)a$. As $c \geq da$ by assumption, we also have $c' \geq na$, so a solution exists to the $n$-dimensional problem. If $c' < bn$, then a solution is trivially $\theta_i = b$ for all $i = 1 \ldots n$. In general, $c' \geq bn$, and we proceed as per the proof of the theorem. Finally, a vector that solves the original $d$-dimensional problem will be given by $(\theta_1, \ldots, \theta_n, a, \ldots, a)$.

**Proof of Theorem 4.2.** Following Theorem 4.1, we need to determine $\alpha^*$ to satisfy the coupling constraint $S(\alpha^*) = c$. Each component $\theta_i$ is a piecewise linear function in the form of a step function with a constant positive slope between the values $a/|w_i|$ and $b/|w_i|$. Let the set $\{\alpha^i\}_{i=1}^{2d}$ be the set of the $2d$ critical points, where the $\alpha^i$ are ordered nondecreasing. The function $S(\alpha)$ is a nondecreasing piecewise linear function with at most $2d$ critical points. We can find $\alpha^*$ by first sorting the points $\{\alpha^i\}$, finding $\alpha^i$ and $\alpha^{i+1}$ such that

$$S(\alpha^i) \leq c \leq S(\alpha^{i+1})$$

by binary search, and then interpolating $\alpha^*$ between the two points. Sorting takes $\mathcal{O}(d \log d)$. Computing $S(\alpha^i)$ at each step of the binary search is $\mathcal{O}(d)$, so $\mathcal{O}(d \log d)$ overall. Given $\alpha^i$ and $\alpha^{i+1}$, interpolating $\alpha^*$ is $\mathcal{O}(1)$, so the algorithm overall is $\mathcal{O}(d \log d)$ as claimed. ∎

**Proof of Theorem 4.3.** Using the infimum formulation of the norm, we solve

$$\min_{x \in \mathbb{R}^d} \inf_{\theta \in \Theta} \left\{ \frac{1}{2} \sum_{i=1}^{d} (x_i - w_i)^2 + \frac{\lambda}{2} \sum_{i=1}^{d} \frac{x_i^2}{\theta_i} \right\}.$$

We can exchange the order of the optimization and solve for $x$ first. The problem is separable and a direct computation yields that $x_i = \frac{\theta_i w_i}{\theta_i + \lambda}$. Discarding a multiplicative factor of $\lambda/2$, and noting that the infimum is a minimum, the problem in $\theta$ becomes

$$\min_{\theta} \left\{ \sum_{i=1}^{d} \frac{w_i^2}{\theta_i + \lambda} : a \le \theta_i \le b, \sum_{i=1}^{d} \theta_i \le c \right\}.$$

This is exactly like problem (17) after the change of variable $\theta_i' = \theta_i + \lambda$. The remaining part of the proof then follows in a similar manner to the proof of Theorem 4.1. ∎