[Reviews · NeurIPS 2014]

Submitted by Assigned_Reviewer_2

In this paper, the authors derive new properties of the k-support norm [1], devise a new efficient algorithm for computing its proximal operator and generalize it to matrices.

First, they show that the k-support norm can be written as a box norm. Box norms are defined using a variational formulation (infimum of quadratics: |w| = \inf_{t \in T} \sum w_i^2 / t_i) and are a special case of the norms described in [2]. Conversely, squared box norm can be written as the sum of squared l2 norm and k-support norm.

Since the box norms are symmetric gauge function, they propose to extend them (and the k-support norm as a special case) to matrices. They also show that the cluster norm is equivalent to spectral box norm. They devise efficient algorithms to compute both the norm and their associated proximal operator, with a complexity of O(d log(d)), where d is the dimension. Finally, they report experimental results for matrix completion problems on synthetic and real datasets, showing that the spectral k-support norm perform slightly better than the trace norm.

This is a solid and well written article about the relations between the k-support norm and the box norms. The extension of those norms to matrices is pretty straightforward. The algorithm proposed in the paper for computing the proximal operator of the k-support norm should be useful to the NIPS community, as it is more efficient than previously known methods: the authors demonstrate that their approach is several order of magnitude faster than the algorithm in [1]. My only small concern is the lack of really convincing experiments about the usefulness of the spectral k-support norm, compared to the trace norm or the matrix elastic net.

[1] A. Argyriou, R. Foygel, N. Srebro. Sparse Prediction with the k-Support Norm. 2012

[2] C. Micchelli, J. Morales, M. Pontil. Regularizers for Structured Sparsity. 2010.
Summary: This is a well written paper, showing how the k-support norm and the cluster norm are related to box norms. This allow the authors to derive a new efficient algorithm for computing the proximal operator of the k-support norm and generalize this norm to matrices.

Submitted by Assigned_Reviewer_13

This paper studies a family of box-norms that include the k-support norm as a special case. These norms can be naturally extended from the vector to the matrix setting. The authors present several interesting properties of the box-norms. Specially, the authors provide a connection between the k-support norm and the cluster norm, and showed that the cluster norm is a perturbation of the spectral k-support norm. They also show that the norm and its proximity operator can be computed efficiently.

The presented theoretical analysis is very interesting. My only concern is its practical significance. Their experiments show that the spectral box-norm and k-support norm achieve similar performance. Thus the spectral k-support norm may be preferred due to its simplicity.
Summary: This paper studies a family of box-norms that include the k-support norm as a special case. The presented theoretical analysis is very interesting. My only concern is its practical significance.

Submitted by Assigned_Reviewer_28

The authors propose a spectral extension of the k-support norm used in the case of sparse regularization in vectors.

The paper also discusses interesting connections and links with other types of spectral norms ( the computational aspects) used in different learning tasks for matrices. Overall this is an interesting paper.

My main concern is that the practical merits of this method is not clear. The experimental section does not seem to suggest that the method is better than counterparts/competitors.
Summary: Paper presents an interesting spectral generalization of a sparsity type norm from the regression/vector case.
Practical benefits of the method are in question.
Author Feedback
Author rebuttal: We thank the reviewers for their helpful comments. All reviewers seemed to acknowledge the theoretical results, however raised some concerns about the practical merits of the paper.

In all experiments that we conducted, including four real data sets, the performance of the spectral k-support and the box norm compared to the trace and elastic net showed consistent improvement. While we acknowledge that the improvement was incremental (on the order of 1%), we verified that this was statistically significant using a t-test (p < 0.001 for the real data sets and p < 0.01 for the simulated data sets).

Furthermore our improved computation of the proximity operator has practical merits as it enables experiments with the vector norms to be performed on datasets of larger dimension.